# The Influence of Social Media on Alcohol Consumption of Mothers of Children and Adolescents: A Scoping Review of the Literature

**Emilene Reisdorfer** [1,*], **Maryam Nesari** [2], **Kari Krell** [1], **Sharon Johnston** [2], **Randi Ziorio Dunlop** [2], **Andrea Chute** [3], **Fernanda dos Santos Nogueira de Goes** [4] and **Inder Singh** [5]

1   Department of Professional Nursing and Allied Health, Faculty of Nursing, MacEwan University, Edmonton, AB T5J 4S2, Canada; krellk@macewan.ca
2   Department of Human Health and Science, Faculty of Nursing, MacEwan University, Edmonton, AB T5J 4S2, Canada; nesarim@macewan.ca (M.N.); johnstons@macewan.ca (S.J.)
3   Department of Nursing Foundations, Faculty of Nursing, MacEwan University, Edmonton, AB T5J 4S2, Canada; chutea5@macewan.ca
4   Centre for Teaching and Learning, MacEwan University, Edmonton, AB T5J 4S2, Canada; santosnogueiradegoesf@macewan.ca
5   Office of Research Services, MacEwan University, Edmonton, AB T5J 4S2, Canada; singhi29@macewan.ca
*   Correspondence: reisdorfere@macewan.ca; Tel.: +1-780-497-5824

**Abstract:** Alcohol misuse is a common problem in many countries, where alcohol is often portrayed as a fun and interactive coping strategy for mothers to manage the demands of motherhood. Social media platforms have established themselves as a popular forum for mothers to share information and create an environment in which mothers may be exposed to and influenced by alcohol-related content. Given the increased social acceptance and normalization of drinking among mothers, especially during the recent pandemic, a critical analysis of social media influences on alcohol behaviours and consumption is warranted. A scoping review mapped the evidence on social media influences and alcohol consumption among mothers of children and teenagers younger than eighteen years old. Several databases were consulted, and the evidence was collated into two themes and seven subthemes. Factors related to alcohol consumption in motherhood include (1) community and social support, (2) coping and mental health, (3) motherhood expectations and identity, (4) alcohol consumption, (5) marketing strategies, (6) everyday issues, and (7) social media influence. Numerous social, economic, and health problems are associated with alcohol misuse. The current literature suggests that social media is a powerful tool to disseminate messages about alcohol and normalize mothers' drinking behaviours.

**Keywords:** mothers; alcohol drinking; social media; review

## 1. Introduction

In Canada, alcohol is the highest consumed psychoactive substance [1]. While men overall consume more alcohol than women, since 2013 there has been a significant increase in women consuming alcohol [2]. Over the last 10 years there has been an 85% increase in women diagnosed with an alcohol-use disorder [3]. According to the Centers for Disease Control and Prevention (2022), 18% of women ages 18–44 report binge drinking [4].

The current literature identifies that social isolation, lack of connection, frustration, and increased stress influence alcohol consumption in women and mothers of young children and adolescents. Additionally, the roles that women take on in society impact their stress levels. Stressors among women, especially mothers of young children and adolescents, have been exacerbated during the past several years of the COVID-19 pandemic due to existing gender inequalities [5]. This inequity in gender roles may lead women to consume alcohol to cope with their stressors and diminish negative effects [6,7]. A study conducted

by Racine and colleagues identified that pre-pandemic maternal anxiety levels were linked to lower coping strategies, more strained relationships, and greater alcohol consumption during COVID-19 [8] (p. 3).

Alcohol use is often portrayed on social media as a fun and interactive coping strategy for mothers to manage the demands of motherhood. Social media posts from sites such as Instagram and TikTok include comments and memes that present alcohol in an entertaining way to normalize alcohol consumption among mothers of young children. Crawford and colleagues noted that social media platforms resonate strongly with followers and can encourage mothers' alcohol consumption [9] (p. 122). Society often judges women for their alcohol use, and social media is where mothers go to find a community of acceptance and social support [6]. However, these sites normalize and market alcohol consumption and may lead to higher-risk drinking [9] (p. 121).

Understanding the effect of social media use on the patterns of alcohol consumption of mothers of young children is needed to develop policies, programs, and strategies tailored specifically to the needs of this population. Given the increased social acceptance and normalization of drinking among mothers, a critical analysis of social media influences on alcohol behaviours and consumption is warranted.

Furthermore, limited information is available about the effects of social media use on mothers' alcohol consumption. The purpose of this scoping review of the literature was to describe what is known from the literature about how social media influences alcohol consumption among mothers of children younger than eighteen years of age.

## 2. Materials and Methods

We conducted a scoping review of the literature to map and identify the gaps in the literature regarding the influence of social media on alcohol consumption among mothers of children younger than eighteen years of age. A comprehensive search of the current literature was undertaken, and a review was not found that clearly answers the proposed research question. Arksey and O'Malley developed a theoretical framework to conduct scoping reviews that includes the following steps: identifying the research question, identifying relevant studies, study selection, charting the data, and collating, summarizing, and reporting the results. The steps were followed and are outlined in the next sections [10,11].

### 2.1. Identifying Relevant Studies

We initially performed a comprehensive online search intended to scope peer-reviewed articles in January 2023. The databases searched were PsycINFO, CINAHL, SocINDEX, Medline, and Academic Search Complete. A combination of the following terms was used to conduct the search: mothers, alcohol drinking, social media. Related keywords were identified, and subject headings were selected according to the databases. The search strategy is summarized in Table 1.

**Table 1.** Search strategy.

| Database | Mothers | Alcohol Use | Social Media |
|---|---|---|---|
| Keywords (common for all databases) | mom OR mommy OR mother* OR matern* OR mummy | "alcohol drinking patterns" OR "alcohol drinking attitudes" OR "alcohol use" OR "alcohol consumption" OR "wine mom" OR "wine-mom" OR "wine mommy" OR "winemom" OR "#winemom" OR "#sendwine" OR "alcohol abuse" OR "hazardous alcohol use" OR "alcohol consumption" OR "alcohol misuse" OR "problematic alcohol consumption" OR "alcohol addiction" OR "binge drinking" OR "heavy drinking" | social media or internet or Facebook or Instagram or TikTok or Snapchat or Pinterest or Twitter or WhatsApp or web* or "social network" or online or hashtag |

* String of characters to search for all terms that begin with that string; # used within a post on social media to help those who may be interested in the topic to be able to find it when they search for a keyword or particular hashtag.

All search results were deduplicated using Mendeley software and imported for screening into Rayyan, a web tool designed to support researchers working on systematic reviews, scoping reviews, and other knowledge synthesis projects [12].

### 2.2. Study Selection and Eligibility Criteria

This scoping review was reported in accordance with the Preferred Reporting Items for Systematic Reviews and Meta-Analyses Extension for Scoping Reviews (PRISMA-ScR) guidelines [13,14]. Titles and abstracts of the obtained citations were screened in the first step to identify potentially relevant papers. In the second step, the full text of the selected articles was assessed further, based on a structured inclusion and exclusion criteria form. Two reviewers performed the two steps of the selection process independently (SJ and IS); discrepancies were resolved either by consensus or by discussing the issue with the third person (FSNG). We manually searched the reference lists of all included papers and relevant journals to find additional citations. The reference lists of the newly identified articles were also examined.

The inclusion criteria included original articles, commentaries, and editorials. The main concept investigated was alcohol consumption, the context was social media influence, and the population of interest was mothers having children up to 18 years of age [15]. Therefore, studies that presented data on the influence of social media on the alcohol consumption of mothers having children up to 18 years of age were included in this study [15]. Studies published in English, Spanish, or Portuguese were considered for further screening. Time limits were not used due to the limited number of articles published and the novelty of the topic. Dissertations, book chapters, and organizational reports were excluded from the search. Grey literature was not considered due to the novelty of the topic and the goal to discover the state of peer-reviewed and scientific knowledge and interest in this field. The inclusion and exclusion criteria were discussed after the titles of the first 100 papers were screened, and the authors decided to maintain the same criteria [10].

Furthermore, because the goal of this scoping review was to map the evidence on social media influences and alcohol consumption among mothers of children and teenagers younger than eighteen years old, the risk of bias assessment was not performed, in accordance with the JBI methodology and the current literature [13,15,16]. As a result, practice recommendations were given with caution [17].

### 2.3. Charting the Data

We used a researcher-constructed data extraction form to extract data from the included articles. Information about the following topics was collected from the included papers: study purpose/context, discipline/country, study methodology, participants, type of social media discussed, and main results. Two researchers from our team extracted the data independently; inconsistencies between the data each extracted were resolved by reviewing the full-text articles together.

### 2.4. Collating, Summarizing, and Reporting the Results

Thematic analysis methodology was used to identify and synthesize themes related to the research questions [18]. A thematic analysis focuses on examining themes within a topic by identifying, analyzing, and reporting patterns (themes) within the research topic.

The thematic analysis procedure was conducted using the software NVivo [19] (released in March 2020) through the following steps: first, authors familiarized themselves with the data and highlighted text directly answering the research questions; a higher-level analysis then followed when the preliminary codes were grouped into themes [20].

## 3. Results

The search strategy yielded 450 papers identified via databases. The list of deduplicated 326 papers was then imported into the Rayyan web-based software [12], where a collaborative screening process of the references list was conducted for eligibility. Three

reviewers (SJ, IS, and FSNG) piloted the inclusion and exclusion criteria on 10% of the included papers. After the pilot, the authors maintained the inclusion and exclusion criteria. SJ and IS screened the papers' titles and abstracts, and FSNG decided on the "maybe" articles. This step eliminated 311 papers, and 15 were included in the full-text screening. A total of four papers were excluded due to methodological or thematic issues, seven were excluded due to wrong population, and one paper was excluded as it was not published in the included languages, resulting in three papers. After revising the reference lists of these papers, four additional studies were added. Finally, seven papers were selected for data extraction and analysis.

More details about the screening process can be found in the PRISMA flow diagram (Figure 1) [13,14].

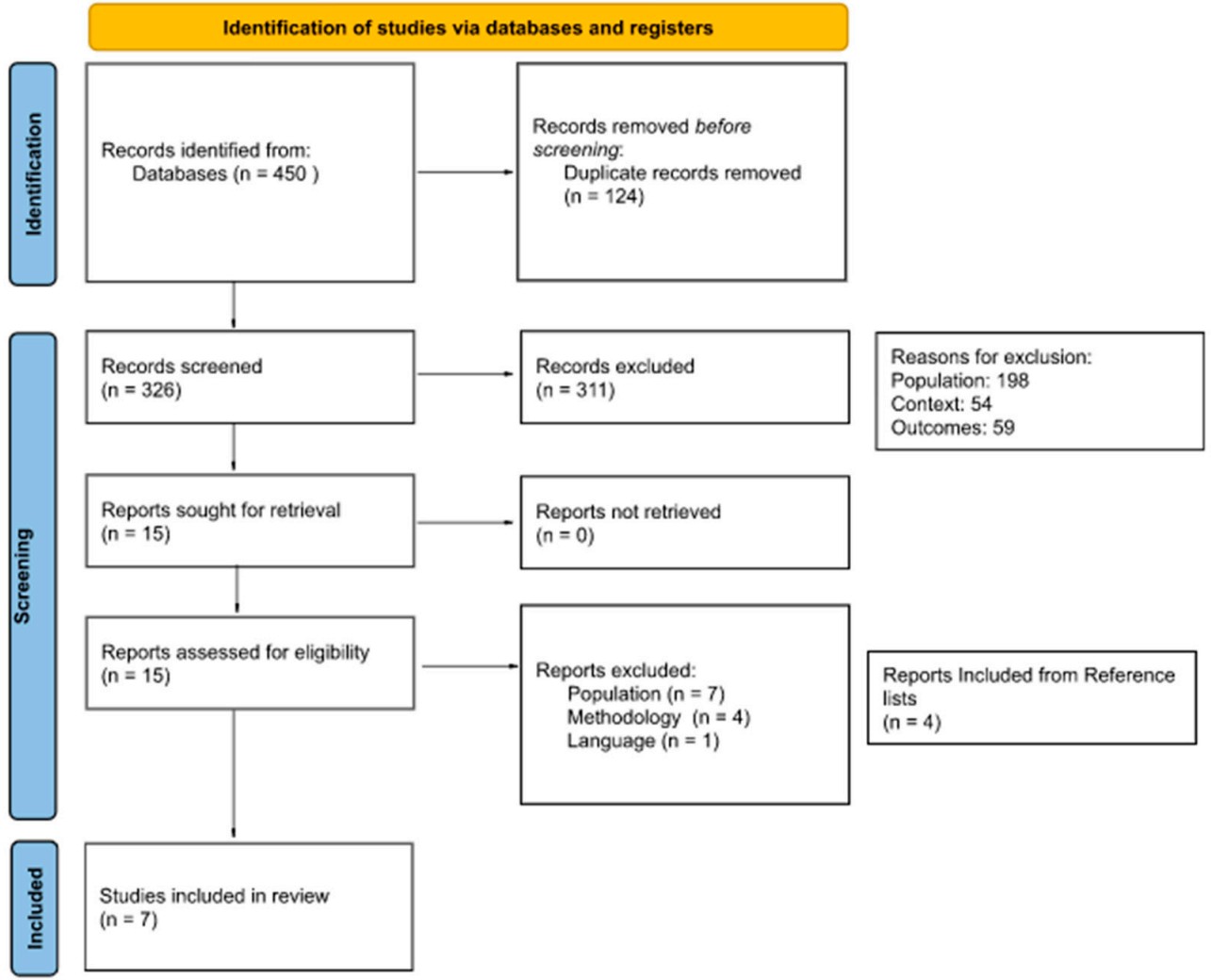

**Figure 1.** PRISMA flow diagram.

Table 2 presents the main characteristics of the seven articles included, including year of publication, country, context, purpose, methods, and type of social media investigated. Overall, the seven included papers were published between 2020 and 2022. Five of the studies were conducted in the United States [9,21–24], and two were conducted in Canada [25,26]. All the papers investigated the influence of social media on the alcohol consumption of mothers (women). The studies either utilized a qualitative approach [9,22–24,26] or were published as commentaries/editorials [21,25].

**Table 2.** Study characteristics.

| Author, Year | Country | Context | Purpose | Methods | Type of Social Media |
|---|---|---|---|---|---|
| Adams, 2022 [21] | United States of America | Behavioural health | To explore the influence of "wine-mom" culture on alcohol consumption and mothers' drinking behaviours. | Editorial. Qualitative overview of the influence of "wine-mom" culture on alcohol consumption among mothers. | Facebook, Instagram |
| Basch et al., 2021 [22] | United States of America | Public health psychology | To describe and analyze social media posts focused on drinking among mothers. | Qualitative study. Content analysis of 100 Instagram posts using the #momjuice and #winemom hashtags. | Instagram |
| Bosma et al., 2022 [25] | Canada | Addictions and mental health | To explore alcohol consumption and the challenges of parenthood. | Commentary. Qualitative overview of how drinking sites offer real support for women experiencing motherhood. | Facebook, online events, websites |
| Crawford 2020 [9] | United States of America | Communication and advertising | To explore the connection between alcohol use, social media, and cultural expectations. | Qualitative study. Thematic analysis of Mommy Drinks Wine and Swears (MDWAS) Facebook page posts. This study analyzed all the alcohol-focused MDWAS posts (99 in total) covering a 1-year period from August 2017 to July 2018. | Social media pages |
| Harding et al., 2021 [26] | Canada | Child and youth health | To explore wine mom culture and ideologies of "good" and "bad" motherhood in online gendered spaces. | Qualitative study. Representative theoretical sample of images and posts. Instagram posts covering a 2-year timespan (2018–2020) and using the hashtags #winemom, #sendwine, #mommyjuice, and #youcansipwithus were initially reviewed to establish what types of posts and content were related to #winemom culture on Instagram. 40 images: 20 posts involving mothers alone with wine and 20 posts involving mothers with wine and children. | Instagram |
| Newman and Nelson, 2021 [23] | United States of America | Sociology and gender studies | To explore "wine mom" discourse and related messaging and its impact on the well-being of women. | Review of the literature. Qualitative overview of how the "wine mom" discourse encompasses a harmful form of conformity to the strain of cultural goals and a lack of institutionalized supports for American mothers. | Online communities and businesses, Instagram |
| Seaver, 2020 [24] | United States of America | History and folklore | To explore internet wine-humour memes and their influence on mental health issues and alcohol use disorders among American mothers. | Review of the literature. Qualitative overview of the role that the Internet has played in reinforcing the popularity of wine as the putative drink of choice for American mothers. | Internet, Facebook, Instagram |

# Used within a post on social media to help those who may be interested in the topic to be able to find it when they search for a keyword or particular hashtag.

Two main themes were identified in the articles reflecting alcohol consumption and the influences of social media on motherhood and are described below. The thematic analysis identified the following themes and subthemes, further described below and shown in Table 3: "Modern motherhood": (1) community and social support, (2) coping and mental health, and (3) motherhood expectations and identity; and "Normalizing alcohol consumption": (1) alcohol consumption, (2) marketing strategies, (3) everyday issues, and (4) social media influence.

**Table 3.** Themes and subthemes identified in the seven articles.

| Modern Motherhood | Normalizing Alcohol Consumption |
|---|---|
| Community and social support [9,21,22,24–26] | Alcohol consumption [9,22–26] |
| Coping and mental health [9,21–26] | Marketing strategies [21–26] |
| Motherhood expectations and Identity [9,23–26] | Social acceptance and normalization [9,21–26] |
| | Social media influence [9,21,22,25,26] |

*3.1. Modern Motherhood*

The theme "Modern motherhood" was present in all seven articles. These articles described the value, in terms of helping with reducing stress or coping, that the mothers ascribed to virtual communities—for example, feeling supported to thrive in the role of mother [9]. Women often feel overwhelmed by the unrealistic expectations placed upon mothers, and being able to parent, work, and drink alcohol proves that they are strong and capable [9,22,24,26]. Creating a cultural space for drinking provided women with an escape, demonstrated resistance to traditional expectations of motherhood, and was perceived as authentic relief from changes in roles and perceptions of mental health [9,23,26].

Motherhood was described as a typical gendered role for women [25]. Impossible demands and unrealistic expectations associated with being a mother have women seeking alcohol consumption and participation in online communities to endure the pressures of motherhood [23].

3.1.1. Community and Social Support

Social media platforms are a common forum for disseminating messages and are popular among mothers for sharing information and coping with the stresses of motherhood [21]. Social isolation during the pandemic played a role in the use of social media as a tool for connection and bonding [21,23,26]. For example, Adams and colleagues identified that the COVID-19 pandemic created "unprecedented pressures on parents and mothers" and "virtual communities often help mothers bond and commiserate" [21]. Harding and colleagues agreed that many mothers rely on remote connection; however, they also found that "women did use the platform to create content that demonstrated that despite the sense of community, these women often felt a sense of isolation, aloneness, stress, anger and frustration" [26].

Wine is utilized as a bonding strategy between people undergoing similar challenges related to motherhood; it is something the women have in common and that facilitates communication. Crawford and colleagues analyzed a popular social media page titled Mommy Drinks Wine and Swears (MDWAS) [9]. They suggested that "participating in a group on a Facebook page is a social behaviour in which drinking returns the MDWAS group members to their youth and helps them feel attractive again." The members are given the gift of transformation, in which "they are transformed into their younger carefree selves, liberated from the stress of work and parenting".

Most of the included papers mentioned social support as an important aspect of the use of social media by mothers these days [9,21,22,24–26]. Women tend to seek social support online due to a lack of community connection and difficulties adjusting to their new role.

3.1.2. Coping and Mental Health

Many women are struggling with mental health issues, and alcohol is often used as a coping strategy to deal with the stress and challenges related to motherhood. Individuals

utilize social media platforms to share their experiences with alcohol use and to receive support during difficult times with their families and children [9,21,23–26].

Harding and colleagues suggested that wine mom culture and online posts with the hashtag #winemom reflected how "many of these women were experiencing poor mental health and were looking for alternative ways to cope, either through alcohol consumption or community" [26] (p. 5). Furthermore, they found that "women are feeling unable to cope or succeed, as barely getting by, and as continually struggling to manage their own mental health, and problematically frame alcohol as the solution" [26] (p. 5).

Newman and Nelson found that "discourse amplifying the necessity of alcohol consumption to endure the demands of motherhood was viral across social media and in advertising and products well before the emergence of COVID-19 and became more prevalent during" [23]. Similarly, Seaver suggested that "American mothers are quietly self-medicating with alcohol to get through each day" [24]. Internet wine memes "help to normalize excessive consumption and high-risk drinking behaviours by rebranding them as coping mechanisms and self-care for women overwhelmed and exhausted by the pressures of modern motherhood" [24]. Bosma and colleagues expanded and suggested that "these events send a message that alcohol is essential to solving problems, dealing with stress, entertaining, and socializing" [25].

### 3.1.3. Motherhood Expectations and Identity

Becoming a mother is a life-changing event that can bring various feelings and challenges. As women transition into their new role as a parent, they develop a new identity, and there is an unrealistic societal expectation that all women should feel joy and happiness [26]. Authors point to unreasonable and unattainable images of motherhood, creating intensive mothering standards that do not accurately depict the realities of motherhood [9]. The struggle to be a "supermom" and the "inability to achieve these expectations results in mothers wrestling with feelings of failure and strain" [23]. Drinking wine within wine-mom culture abates these intensive standards and traditional expectations of motherhood. However, dichotomizing mothers as either "good" moms or "bad" moms may be seen to reinforce the myth of motherhood:

> "The hashtag #winemom has created a cultural space that allows women to form a community that enabled reinforcing feedback that mothers remained 'good mothers' and were not descending into 'bad' mother territory". [26] (p. 5)

Alcohol consumption brings support from and bonding opportunities with other women experiencing similar life events [9]. It allows women to feel empowered and competent in their modern motherhood skills [9,26]. Moreover, the wine-mom discourse provides an escape from domestic and family obligations and an opportunity for authenticity, affirmation, and self-care development [23,26]. For example: "real moms drink and they don't judge—unless they are judging those who are not 'authentic drinkers'" [9].

The wine-mom culture projects a lifestyle and visually represents women as healthy, fit, and trendsetting [26]. Modern mothers also see alcohol use as a means of empowerment:

> "Empowerment expressed by young mothers in relation to their alcohol use where women discussed their consumption as a means to take back their pre-child identity and to assert the equality and independence in contrast to their (male) partners". [26] (p. 5)

Authors found that "wine is given a special reverence in the group, perhaps because of its connection to the group's identity" [9]. Conversely, this gendered identity and resistance to traditional expectations, Newman and Nelson argue, "can be understood as a 'false' resistance." They suggest that "wine mom discourse facilitates (White, middle-class) mother's conformity to the hegemonic strains of motherhood via alcohol consumption" [23].

### 3.2. *Normalizing Alcohol Consumption*

The second theme, "Normalizing alcohol consumption", also emerged from the seven articles. Social media messages helped to influence excessive alcohol consumption and fostered widespread acceptance of online content that supported social connection as a way of coping with the stresses of motherhood [9,21–26].

Social media is a powerful tool for disseminating messages about alcohol [9,21]. Marketing strategies that target women create social influence to normalize mothers' drinking behaviours [21,22,24–26].

#### 3.2.1. Alcohol Consumption

Social media platforms are used to find social connection and support, and a positive lens often characterizes alcohol use [9,22–25]. Studies that described alcohol-related content posted by mothers on Instagram during the COVID-19 pandemic found that "the posts depict alcohol consumption as risk-free and minimize its negative consequences" [22]. Researchers argue that "'Team Mommy' and its followers feel safe drinking large quantities" and "the potential risks are mitigated by the fact that the mommies drink at home or with other mommies" [9]. Furthermore, rituals associated with alcohol consumption are shared and encouraged via social media posts [24,25]. Crawford and colleagues found in their study of Mommy Drinks Wine and Swears (MDWAS) that one of "the most striking ritual-related elements of the posts were the visual prominence and positive representation of alcoholic beverages" [9]. Only a handful of posts studied discouraged alcohol consumption or presented disclaimers [22].

#### 3.2.2. Marketing Strategies

The alcohol industry utilizes several strategies tailored to different parts of the population. Mothers are specifically targeted as a new group of consumers, and the strategies vary from labelled t-shirts and humorous memes to hosting and encouraging special events at which alcoholic beverages are used to create bonding and friendship [21–26]. Seaver explains that:

> "Predominantly white women of childbearing age, especially those who live in middle-and upper-middle class suburban areas, are the same demographic that is frequently targeted by advertisers marketing wine products to the American public". [24]

Also, the included studies show that the alcohol industry portrays alcohol as a symbol of freedom, a very appealing strategy for this population [25]. Alcohol companies promote online "Mummy drinking" events and opportunistically advertise to women, using "mompreneurs" to sell alcohol. This may be a result of the "few regulations on Instagram regarding the promotion of alcohol" [26] (p. 6).

#### 3.2.3. Social Acceptance and Normalization

Alcohol is a psychoactive substance that has been widely used and normalized by our society for millenniums. It is part of the social fabric and is often consumed in pleasurable gatherings and casual situations. The presence of alcohol on social media platforms is positively associated with humour and bonding; therefore, it contributes to the increased social acceptance and normalization of drinking among mothers [9,22]. Authors found that "most of the commenters joke and brag about losing control and using loss of control as a form of escape" [9].

Another aspect of the normalization of drinking is the use of social media for constant feedback and reassurance [9,21–26]. The pandemic compounded alcohol consumption as people tried to manage changing circumstances. Wine-mom groups and the humorous protective buffer they provided to members may have "provided a relieving counterpoint to the strong negative emotions felt by many as the pandemic unfolded, lockdowns were mandated, and women in particular faced sudden and dramatic changes in roles and

perceptions of mental health" [22]. This was demonstrated in the way that alcohol-related memes posted in specific social media groups frequently received several "likes" and comments, mostly positive and encouraging [22].

### 3.2.4. Social Media Influence

Social media significantly influences women's lives, emerging as a popular platform for mothers to share the struggles and joys of motherhood [22]. Humorous content often targets women and mothers, depicting the challenges faced in a light and amusing manner [9,22]. Moreover, influencers and "mompreneurs" (a mother who has started her own business [27]) create content that portrays successful women combining drinking with fashionable life activities—such as travel, clothing, and home decor, among others—on top of their family obligations. Harding and colleagues found that:

> "Some women posting content on Instagram also identified themselves as 'wine educators' and 'winepreneurs', that 'got' the struggles of motherhood and wanted to ensure that women were provided with a quality wine product to take the edge off being a parent". [26] (p. 6)

Content creators on Instagram also use the hashtag #winemom to craft textual and visual narratives that promote the image of being a 'supermom' [26]. This created a sense that all women should be able to engage in similar activities if they are to achieve success.

## 4. Discussion

This scoping review of the literature aimed to describe what is known from the literature about how social media influences alcohol consumption among mothers of children younger than eighteen years of age. The results indicate that women with children and/or adolescents who utilize social media experience the considerable effects of the expectations of modern motherhood and the normalization of alcohol consumption.

### 4.1. Modern Motherhood

Social connections give people a sense of overall worth and psychological well-being and access to resources during difficult times and life transitions [28]. Our study identified that women used social media to connect with others going through similar experiences in order to seek social support. A study by Tani and Castagna [29] revealed that social support is a significant protective factor against a long, difficult, and painful childbirth experience. In addition, the study identified that broader social support from families, friends, and partner improved all outcomes related to parenting.

On the same note, Archer and Kao [30] highlighted the importance of Facebook as a mechanism for social support among mothers. The study indicated that the primary motivation for social media use was to maintain contact with family and friends, relieve boredom, keep up with the news, and find information on other people. The participants also revealed that Facebook had negative aspects, such as being addictive, potentially a cause for depression or anxiety, and superficial. In addition, mothers were concerned about how their behaviour could influence their children's use of social media and electronic devices in the future [30].

Despite evidence unveiling the importance of social connections to mothers and children [29–32], social support can be difficult to access. As evidenced by the findings of our study, many women rely on virtual communities as the main source of connection due to the challenges brought to society by the pandemic in the last few years, as well as lack of time and lack of knowledge about possible opportunities. To meet other women going through similar situations, mothers also need safe spaces in which to feel comfortable discussing issues and worries without being judged or criticized [29].

Women who are going through similar motherhood-related issues might use wine to connect with one another and facilitate conversation. Alcohol is a psychoactive substance widely available and accepted by society [30]. It is frequently used as a coping mechanism to deal with the stress and difficulties associated with parenthood, and many women

are battling mental health issues [33]. The use of alcohol as a coping strategy reflects pervasive, damaging societal narratives that suggest alcohol is beneficial during times of loss or trauma, which then affect resilience and recovery [33–35]. Other potential causes of mothers' drinking include perceived social pressure, health concerns not being understood or being discounted, and coping with trauma and mental health issues [33,36]. Furthermore, a study conducted with pregnant women found that many individuals utilize alcohol as a coping mechanism to deal with the difficulties associated with the period of life they are living [35].

Although a common coping strategy, alcohol use can negatively impact parenting and relationships between mothers and their children. Mothers who reported work–family difficulties seem to have a propensity for drinking to deal with the issues and anxieties brought on by these conflicts. When the stress and adverse effects of the work–family conflict are not appropriately managed, individuals are at risk for long-term health effects [37].

Parenting stress is correlated with poorer mental health and less social support, which has an adverse effect on parenting techniques and family interactions, while highly stressed mothers are particularly vulnerable to alcohol misuse [38].

Additionally, drinking is linked to much less consistent family rituals, worse mother–child communication, and lower parental participation [39]. Moreover, alcohol use and abuse are related to an increased risk of physical [40], mental [41,42], and behavioural abuse [43].

Being a mother is a life-changing experience that can bring a range of emotions and difficulties. There is an unreasonable social expectation when women adjust to their new position as parents and build a new sense of self. The integration of conceivable motherly identities is a requirement of motherhood. These mothering selves are shaped by discourses that frame what it means to be a mother and reflect the tensions, inconsistencies, and ambivalence experienced [44].

Mothers are faced with unrealistic societal expectations regarding behaviours and attitudes. They are expected to feel pure joy and happiness when instead they might experience different feelings related to breastfeeding, physical appearance, and activities with their children, for instance. Unrealistic and unreasonable conceptions of motherhood create rigorous mothering norms that do not adequately reflect the realities of parenting. Banister and colleagues [44] pointed out that some women resist what they perceive as normative discourses associated with mothering. Alcohol consumption might serve as a way in which women resist these expectations and attempt to maintain some of their "old selves" and identity. It has also been demonstrated that drinking wine may be used to express defiance of conventional parenting expectations by undermining them via wine consumption. Young mothers spoke of their empowerment in connection to alcohol use, seeing it as a way to reclaim their pre-child identities and establish their equality and independence in contrast to their (male) partners [45].

The reality of the concept of motherhood had a tremendous impact on whatever expectations or preconceived notions people may have had. Women discovered after becoming mothers that this mindset is extremely different from reality. A study conducted by Choi and colleagues [46] demonstrated that, as a result of becoming mothers for the first time, some women realized some facts they had not previously recognized. Motherhood had, for most of the women, either not been or not been at all what they had anticipated [46].

The "good mom, bad mom" phenomenon was also uncovered in this study. Being able to still perform expected activities (being good) even after drinking large amounts of alcohol (being bad) was considered a success, and women felt that they were good moms. Bad moms were seen as stressed, mean, and unable to cope with motherhood expectations. Alcohol was seen as a factor that helped women feel good about themselves. Existing research implies that substance-abusing women appreciate motherhood because it allows them to realign themselves with conventional feminine qualities and feel good about themselves [47].

*4.2. Normalizing Alcohol Consumption*

Normalization of alcohol use refers to presenting a positive portrayal of alcohol as a coping mechanism to reduce stress and of prestigious types of drinks that can ease social conversations. The normalization of alcohol consumption is shaped in numerous ways. It happens, in the first place, by parents exposing their children to alcohol at parties or on TV shows displaying people's drinking habits as a daily practice. Recently, social media has been instrumental in normalizing alcohol use by showing alcohol-related messages from users with the least number of limitations and by facilitating unregulated alcohol marketing from companies [48]. The possibility of sharing user-generated messages and images of any kind on social media has created an atmosphere in which people can be influenced by many other users whom they do not even know. Considering the popularity of these media, it is possible to estimate the magnitude of their effects on people's lives. It is not surprising to hear that people, including overwhelmed young mothers, seek support groups on social media to replace face-to-face interactions and to engage in ritualistic alcohol consumption.

Advertisements on social media seem coercive in nature, as social media users have limited control over receiving the advertisements. Facebook allows advertisers to target users with relevant advertisements based on the users' browsing habits on Google. Since 2012, because of Facebook Exchange Service, visiting a website on Google in search of a product triggers the presence of numerous advertisements for the product on the user's Facebook page [49]. This is known as "remarketing", which benefits companies by increasing their sales; however, it might influence users negatively by tempting them to buy products they do not necessarily need. Like other advertisers on Facebook, alcohol companies want the users to "like" their brands and post pictures while drinking the specific brands. Three beer and two spirits brands were followed by more than 10 million fans on Facebook in 2015. Unlike traditional mass media such as TV, there are limited regulations on controlling algorithms that promote alcohol marketing on social media [49]. A study conducted in Australia had similar results and identified that there were alcohol advertisements posted on average every 35 seconds, with the themes of easy access without leaving home (58%), buy more (35%), drink during COVID-19 (24%), and drink to cope (16%) [50].

A study aimed at analyzing the extent and nature of gendered alcohol marketing content on Facebook and Instagram found out that both new and established methods were being employed to target women and men. Drinking was portrayed as a traditionally feminine activity and a crucial part of "performing". Traditional gender roles and stereotypes were simultaneously reinforced and rejected to encourage alcohol consumption, and women were assigned a variety of gender roles that honoured their distinct pleasures and accomplishments. A significant shift toward appropriating feminist and equality ideas that may appeal to a larger variety of women, especially those embracing feminist identities, was seen, moving away from sexualizing and denigrating women [51].

Moreno and Whitehill [52] discussed the influence of alcohol-related content posted on social media by the users and advertisers on individuals' drinking behaviours. Using social learning theory, they explained how young people learn new behaviours by directly experiencing them, as well as observing them on social media platforms. A systematic review with meta-analysis showed a moderate strength relationship between being exposed to alcohol-related messages on social media and alcohol consumption and consequences in young adults. The effects are not limited to alcohol drinkers. Observing these contents by individuals also increases the likelihood of initiating drinking alcohol [53], which explains the role of social media in normalizing alcohol use.

As stated by Maturkanič et al. [54], people who are a part of certain social groups or communities receive assistance from others. Help can come in many different forms, including doing good deeds for others while also putting one's own life in danger. Human beings have the capacity to make decisions and the desire to assist in these situations [54].

*4.3. Significance for Nursing Practice and Future Directions*

The findings of this scoping review identify themes of the stress and challenges that come along with modern motherhood and the influence of social media on alcohol consumption as a solution. Children do not raise themselves, as much as parents may wish at times that they did, and the mental health and well-being of caregivers needs attention to ensure the overall functioning of the family. As a profession that comes into contact frequently with mothers throughout their child's early life, nurses have an opportunity to create space for conversations about coping and alcohol use. Identifying opportunities to engage mothers in conversations—at public health visits, for example—about the challenges they are facing and the coping skills they are using may help identify those that need further intervention and support for their alcohol use. Continuing these conversations during nursing interactions as the child ages and the mother adapts to their changing identity and societal roles could be important in ensuring that mothers have an outlet outside social media to help them normalize their feelings around motherhood. Through these interactions there are opportunities to discuss healthy coping mechanisms and connections to additional services to support their individual needs. The themes identified in this scoping review increase nurses' awareness of the challenges that mothers are facing and the continued need to explore evidence-informed interventions to best support the health of mothers and their families.

Future research should be conducted to evaluate women's experiences with social media and alcohol consumption, as well as the social media influences on women's drinking habits. Furthermore, studies should focus on diverse populations and settings and analyze primary data.

*4.4. Limitations of the Study*

The results of this study are limited to the evidence collected from the seven studies; among them, one is an editorial, and one is a commentary. The scarcity of evidence on this topic could be due either to the novelty of the issue or to the fact that the power of social media on people's health has not yet been fully understood. We focused on papers published in English, Portuguese, and Spanish; one obtained paper was excluded as it was not published in the included languages. Additionally, the richness of the evidence from this review can be affected by the limitations of the original studies. Drinking habits are a sensitive topic to be discussed specifically with women. This might influence the number of participants in each study and can limit variation in participants. A call for participants for studies on this topic might attract only women from a cultural background in which alcohol consumption by women is accepted. In cultures where drinking alcohol is generally prohibited for women or not expected from them, recruiting female participants on studies into alcohol-related topics would be difficult, if not impossible.

**5. Conclusions**

Despite the scarce amount of research available on how social media influences alcohol consumption among mothers, the current literature seems to suggest that social media is a powerful tool for disseminating messages about alcohol and normalizing mothers' drinking behaviours. Results from qualitative studies and commentaries/editorials indicate that social media influences alcohol consumption among mothers of children younger than eighteen years of age by encouraging the use of virtual communities to get social support to deal with life during the pandemic, their new motherhood identity, and the unreasonable and unattainable expectations of being a mother. Alcohol has been advertised as a way to connect mothers having similar issues and as a coping strategy to deal with their mental health. The paucity of literature on how social media influences mothers' alcohol consumption indicates the serious need to continue investigating this topic and identifying the impact of social media on motherhood. It is equally important to investigate how the healthcare system can support mothers' mental health and health promotion strategies. Finally, a uniform regulatory response and legislation for social media content

moderation are urgently needed to diminish social media's negative impact on mothers' and families' health.

**Author Contributions:** Conceptualization, E.R., M.N., K.K., S.J., R.Z.D., A.C. and F.d.S.N.d.G.; methodology, E.R., M.N. and F.d.S.N.d.G.; software, S.J., F.d.S.N.d.G. and I.S.; validation, E.R., M.N. and K.K.; formal analysis, E.R.; investigation, E.R., S.J. and F.d.S.N.d.G.; resources, E.R.; data curation, E.R., M.N., K.K. and S.J.; writing—original draft preparation, E.R., M.N., K.K., S.J., R.Z.D., A.C., F.d.S.N.d.G. and I.S.; writing—review and editing, E.R., K.K., R.Z.D. and F.d.S.N.d.G.; visualization, E.R., M.N. and K.K.; supervision, E.R. and F.d.S.N.d.G.; project administration, E.R., S.J., A.C. and F.d.S.N.d.G. All authors have read and agreed to the published version of the manuscript.

**Funding:** This research received no external funding.

**Acknowledgments:** The authors would like to acknowledge the support provided by the MacEwan University's Library, especially the librarian Jody Nelson.

**Conflicts of Interest:** The authors declare no conflict of interest.

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
