# Peer review of "The Influence of Social Media on Alcohol Consumption of Mothers of Children and Adolescents: A Scoping Review of the Literature"

_nursrep, doi:10.3390/nursrep13020061_

Round 1
Reviewer 1 Report
Thank you for the opportunity to review this article.
The present study aimed to describe what is known from the literature about how social media influences alcohol.
Introduction:
The authors present the topic by revealing the essential aspects of the understanding of the study. Apresentam o objetivo, no entanto, falta a pertinência da realization da Scoping Review. Does a similar study exist that has been published recently? How can the authors guarantee that this publication does not already exist? Do the authors register the protocol for revision? It is important to do it to avoid duplication of work.
Materials from the method:
The authors will adequately describe the materials and methods, failing to define the PCC method since it is a Scoping Review. I must include this information.
Results: In tabela 2 the authors present the characteristics of the study, but they do not present the results of the same. It is important to put this information on the table for the analysis of the results.
The authors organized the results by themes and subthemes. However, they must be standardized, because there are places where the authors refer to the categories and subcategories. Have you used any theoretical models for this analysis? I must follow him.
Discussion:
The authors compare their results with the available literature. I will adequately identify the limitations of the present study.
Conclusion:
Give answer to the objective of the study.
Author Response
Dear reviewer
We appreciate all your comments and suggestions to improve the quality of our study.
Please find the answers to the review in the attached document.
Kind regards,

Reviewer 2 Report
Dear authors
I appreciate the opportunity to review: The influence of social media on alcohol consumption of mothers of children and adolescents: a scoping review of the literature.
The paper mapped the evidence on social media influences and alcohol consumption among mothers of children and teenagers younger than eighteen years old.
The study has a very clear structure, line and contains very interesting results.
for example: greater alcohol consumption during COVID-19 -
The authors wrote the study with commitment, passion and knowledge. - excellent job.
The article contains all the important components for a scientific article
It is my pleasure to strongly recommend this article for publishing.
I would like to mention that problem of alcoholism is not only in Canada but also (and more problematic ) in Czechia and Slovakia - please add to your discussion :
1.
Maturkanič, P.; Čergeťová, I.T.; Králik, R.; Hlad, Ľ.; Roubalová, M.; Martin, J.G.; Judák, V.; Akimjak, A.; Petrikovičová, L. The Phenomenon of Social and Pastoral Service in Eastern Slovakia and Northwestern Czech Republic during the COVID-19 Pandemic: Comparison of Two Selected Units of Former Czechoslovakia in the Context of the Perspective of Positive Solutions. Int. J. Environ. Res. Public Health 2022, 19, 2480. https://doi.org/10.3390/ijerph19042480
2.
/ Edward Zygmunt Jarmoch et al, 2022. Social work and socio-pathological phenomena in the school environment. In: Acta Missiologica. - ISSN 1337-7515, Vol. 16, n. 2 (2022), pp. 130-145.
Selected socio-pathological phenomena - p. 132
Author Response

(The authors gave the same response as above.)

Reviewer 3 Report
Mothers’, housewives’ drinking has been studying for many years. Also a growing number of literature examines how pandemic and social media have been changing alcohol consumption habits. The novelty of the study is that authors try to reveal the effects of social media on normalization of women’s home alcohol drinking. As a method they conducted a scoping review of the literature which is appropriate for the purpose. Their conclusion is that the social media contributes to the acceptance and normalization of mothers’ alcohol consumption.
In detail:
The introduction chapter gives a proper overview of the literature. The cited references are relevant and published recently. This chapter well supports the aim of the authors, and this aim is clearly formulated.
They systematically describe the study method. The search strategy, selection and eligibility criteria are correctly presented. I can’t clearly understand why only 7 papers were selected into the final analysis, when they could identify 450 papers by the search strategy. The different steps are properly presented (also by a diagram) but it has not been properly clarified why this high number of articles was excluded from the analysis. Here I miss some additional explanation of the reasons of exclusion. Tables well summarize the characteristics of the seven included articles.
The discussion chapter only partly refers to the results of the study. Here the authors present all the thoughts they have in relation with the problem. I suggest more focusing on the study related conclusions than discussing all related issues.
To sum up, the main contribution of the paper is that it points out the important effect of social media on normalizing the alcohol use during motherhood. Proper research design and correctly presented methodology are a strong point of the article.
I miss some explanation for the reasons of excluding a high number of articles and a more focused conclusion chapter.
Author Response

(The authors gave the same response as above.)
